# Flavonoids Targeting Cancer Stem Cells for Augmenting Cancer Therapeutics

**DOI:** 10.3390/ijms222313044

**Published:** 2021-12-02

**Authors:** Ari Meerson, Soliman Khatib, Jamal Mahajna

**Affiliations:** 1Department of Natural Products and Nutrition, MIGAL—Galilee Research Institute, Kiryat Shmona 11016, Israel; arim@migal.org.il (A.M.); solimankh@migal.org.il (S.K.); 2Faculty of Sciences, Tel Hai Academic College, Qiryat Shemona 12208, Israel

**Keywords:** natural products, cancer stem cells, cancer therapy, flavonoids

## Abstract

Cancer stem cells (CSC) have been identified in several types of solid tumors. In some cases, CSC may be the source of all the tumor cells, the cause of the tumor’s resistance to chemotherapeutic agents, and the source of metastatic cells. Thus, a combination therapy targeting non-CSC tumor cells as well as specifically targeting CSCs holds the potential to be highly effective. Natural products (NPs) have been a historically rich source of biologically active compounds and are known for their ability to influence multiple signaling pathways simultaneously with negligible side effects. In this review, we discuss the potential of NPs in targeting multiple signaling pathways in CSC and their potential to augment the efficacy of standard cancer therapy. Specifically, we focus on the anti-CSC activities of flavonoids, FDA-approved drugs originating from natural sources. Additionally, we emphasize the potential of NPs in targeting microRNA-mediated signaling, given the roles of microRNA in the maintenance of the CSC phenotype.

## 1. Introduction

In recent years, modern cancer therapies have made significant progress in the treatment of many solid tumors. However, cancer recurrence is inevitable in most cancers due, in part, to the existence of a small subset of tumor cells that are refractory to therapy, called cancer stem cells (CSCs). CSCs are described as belonging to a group of tumor-initiating cells (TICs) that possess stem-like characteristics. The role of CSCs in tumor formation was first described by Bonnet and Dick in the late 1990s [1], and it is evident that CSCs play a crucial role in tumor development and recurrence for many cancer types [2,3,4,5]. Traditional treatments, such as chemotherapy and radiation, were originally developed to kill the rapidly dividing bulk population of cells within the tumor. However, whereas these therapies can shrink the tumor, the effects are often transient, and recurrence remains a reality for a substantial proportion of cancer patients. Despite eradicating the bulk of the tumor mass, these treatments have proven insufficient in eliminating CSC populations. For example, CD133+ glioma CSCs have been shown to resist radiation therapy to a higher degree than their CD133- counterparts [6]. Breast CSCs exhibit resistance to radiotherapy in addition to common chemotherapy treatments [7]. Thus, the remaining CSCs following standard cancer therapy eventually form new, fully developed tumors. Complete eradication and prevention of cancer relapse require the removal of the stem cell subpopulation within a tumor. A combination therapy of drugs that target non-CSC tumor cells with therapies that specially target CSCs is likely to be the most efficient regime in cancer therapy (Figure 1).

## 2. Hallmarks of CSC

Cancer stem cells (CSCs) have been identified in several types of solid tumors, including breast cancer, brain tumors, lung cancer, colon cancer, and melanoma. Cancer stem cells have the capacity to self-renew, to give rise to progeny that is different from them, and to utilize common signaling pathways. Cancer stem cells may be the source of all the tumor cells present in a malignant tumor, the cause for the resistance to the chemotherapeutic agent used to treat the malignant tumor, and the source of cells that give rise to distant metastases. It has become increasingly evident that the CSC characteristic varies between cancer types and even in individual patient tumors of the same subtype. Below are several hallmarks shared by many CSC in different tumor types.

**Self-renewal:** CSCs are a population of self-renewal cells with high tumorigenic potency. Self-renewal is defined as the ability to go through numerous cycles of cell division, while maintaining the undifferentiated state. This requires cell cycle control and often maintenance of multi-potency or pluripotency. Self-renewal programs involve networks that balance proto-oncogenes (promoting self-renewal), gate-keeping tumor suppressors (limiting self-renewal), and care-taking tumor suppressors (maintaining genomic integrity). In some stem cells, the process of self-renewal is de-regulated, resulting in the expansion of these cells and in tumors. WNT/β-catenin, transforming growth factor-β, Hedgehog, and Notch are important signals for maintaining self-renewal in CSCs.

**EMT capability:** Epithelial–mesenchymal transition (EMT) is the process undergone by epithelial cells in which the cells alter their morphology, lose their polarity, and break cell–cell or cell–matrix adhesions. In this way, the cells gain mobility and invasive potential. CSCs are hypothesized to possess enhanced EMT capability, enabling the cells to survive in the absence of cellular adhesion in addition to enhancing their resistance to apoptosis. CSCs which have undergone EMT are thought to then reattach and produce metastatic tumors or circulate throughout the body in a dormant state, only to become active years later and cause distant cancer relapse to occur [8].

**Drug resistance:** Drug resistance is commonly associated with CSC populations. Drug resistance in CSCs is mediated in part by ATP-binding cassette (ABC) transporters that are overexpressed in CSCs [9]. ABC transporters are transmembrane proteins that serve a crucial cytoprotective role for healthy stem cells throughout the body by removing toxic compounds from inside the cells. Members of the ABC transporter family that appear to be highly expressed in CSCs include ABCB1, ABCG2, and ABCB5 [10]. ABCB1 contributes to the efflux of many widely used chemotherapeutic agents. Reduction in the expression of ABCB1 has been shown to lead to an increased chemotherapy sensitivity of colorectal CSCs. By targeting ABC transporters, the unique resistance of CSCs can theoretically be reversed, sensitizing them to traditional chemotherapy treatments [11].

**Tumorsphere formation:** EMT capability is assessed by removing any opportunity for cellular attachment. When in these conditions, cells without EMT capability will die, leaving only cells that have undergone the transition. The remaining cells often grow in what is referred to as tumorspheres, which are enriched in CSCs in numerous tissues [12,13].

CSCs utilize many of the pro-survival signaling cascades and maintenance proteins seen in healthy stem cells. CSCs tend to survive cellular stresses capable of eliminating differentiated cancer cells, similarly to non-malignant stem cells. For example, mechanistic targets of rapamycin (mTOR) and signal transducer and activator of transcription 3 (STAT3) play a role in the maintenance and proliferation of healthy and cancer stem cells [14]. The stem cell maintenance proteins Wnt, Hedgehog, and Notch are also upregulated in CSCs. These molecules play a major role in maintaining the stemness of CSCs and activating the expression of stem cell-related transcription factors such as octamer-binding transcription factor (Oct4) and Nanog as well as influencing EMT [15]. Moreover, involvement of Hedgehog (Hh) signaling in CSC has been suggested in studies of multiple human cancers [15]. Hh signaling is activated in Bcr-Abl-positive leukemia stem cells (LSC), and pharmacological inhibitors of Smo reduced leukemia stem cells in vivo [16], suggesting that Smo inhibition could be an effective treatment strategy in reducing tumor relapse and drug resistance in chronic myeloid leukemia. Dysregulation of these pathways is hypothesized to promote gradual CSC differentiation, leading to decreased tumor viability in response to chemotherapeutics, and making them an attractive target for the treatment of both bulk tumors and CSCs [17].

## 3. CSC Markers

A number of biomarkers were found to be overexpressed on CSCs of most solid tumors such as CD44 (hyaluronate receptor), CD133 (prominin-1), EpCAM (epithelial cell adhesion), ALDH (aldehyde dehydrogenase 1A1) and ABCG2 (ATP-Binding Cassette Sub-family G Member 2). However, other markers were overexpressed on CSCs of a subset of tumor types such as CD117 (c-kit) in glioblastoma, ovarian and lung cancers [18,19,20]; CD24 (Heat-Stable Antigen) in pancreatic cancer [21]; CXCR4 (chemokine receptor) in pancreatic and breast cancers [21,22]; CD90 (thymocyte differentiation) in breast and colon cancers [22,23], CD49f (Integrin α6) in prostate cancer [24,25] (Table 1). Interestingly, leukemia stem cells (LCS) are positive for CD34 (hematopoietic progenitor cell antigen) and negative for CD38 (cyclic ADP ribose hydrolase) [26].

**CD44-positive:** Cluster of differentiation 44 (CD44) proteins are integral membrane glycoproteins that play a role in cell attachment to the extracellular matrix by binding to hyaluronan (HA). The expression of CD44 has been used as a putative marker for cancer stem cells [27]. CD44 regulates the growth, migration, and invasion characteristic of CSCs in addition to modifying the extracellular matrix of tissues to support new tumor formation [8,27]. CD24 is another cell surface protein and a heat-stable antigen; its expression is a marker for predicting clinical outcome and the expression of other stemness-related genes, in conjunction with CD44 [28].

**CD133-positive:** Cluster of differentiation 133 (CD133) is a pentaspan membrane surface protein that is also commonly used as an indicator of CSCs in many cancer types [8]. CD133 expression has been positively correlated with poor outcomes for cancer patients [8].

**ESA or epCAM:** Epithelial specific antigen (ESA), also known as epithelial cell adhesion molecule (epCAM) is a surface marker typically expressed on epithelial cells and has been used to identify CSCs in many types of cancer [29]. ESA regulates cell-to-cell adhesion, migration, and invasion of cancer cells [29].

**Side population:** Although characteristics of CSC seem to vary from one tumor type to another, nevertheless several key characteristics are shared by most cancer types. Hoechst 33342 is a stain capable of permeating intact cell membranes, which produces blue fluorescence when bound to nuclear DNA. Hoechst 33342 excluding cells, also known as the side population (SP), of tumors have been investigated as a source of drug-resistant CSCs [11]. The ability of the SP to exclude Hoechst 33342 is a result of ABC transporters, specifically ABCG2, make SP isolation an indirect method of CSC isolation based upon ABC transporter expression [10].

**ALDH activity:** Aldehyde dehydrogenase (ALDH) activity has been used to identify CSCs [30]. ALDH catalyzes the oxidation of aldehydes entering or produced within the body. By oxidizing aldehydes, these enzymes transform potentially deleterious compounds into carboxylic acids, preparing them for cellular metabolism and thus contributing to the protection of CSCs from chemotherapy treatments. By eliminating ALDH activity from tumors, the breakdown of chemotherapeutic agents within the tumor can be slowed, resulting in more effective treatment. Cytotoxic compounds that do not act as substrates for ALDH enzymes or that reduce their activity may have a unique ability to induce apoptosis in CSCs and act as more effective long-term treatments.

**Table 1 ijms-22-13044-t001:** CSC markers of the major cancer types.

Cancer Type	CSC Markers											Reference
	CD44	CD133	CD117	CD24	EpCAM	CXCR4	ALDH	ABCB5	ABCG2	CD13	CD90	
Ovarian Cancer	+	+	+				+		+			[18]
Breast cancer	+			-	+	+	+		+			[22]
Brain tumor		+					+				+	[31]
Pancreatic cancer	+	+		+		+	+		+			[21]
Colon Cancer	+	+			+		+					[23]
Liver cancer	+	+			+				+	+	+	[32]
Prostate cancer	+	+					+					[33]
Lung Cancer		+	+		+		+		+			[19]
Glioblastoma	+	+										[20]
Melanoma	+	+	+				+	+				[34]

The generation of CSCs is attributed to the tumor microenvironment [35]. Consistently, it has been shown that the tumor microenvironment contributes to cell plasticity during tumor development, initiating stem-like programs in non-CSC or normal cells and involving key signals, such as the Wnt pathway [36] or inflammation-associated signatures [37]. Micro-environmental cells can largely affect the maintenance and behavior of CSCs by regulating metabolic changes and the secretion of growth factors/cytokines. Tumor microenvironment affects the induction of the epithelial–mesenchymal transition (EMT) phenotype necessary for CSC generation, as well as increases in the number and renewal potential of CSCs and induction of pluripotency-associated transcription factors, such as Oct-3/4, Nanog and Sox-2 [38,39]. Interestingly, CSCs seem to have unique immune evasion features, including overexpression of PD-1/PD-L1 molecules [40,41], suggesting the potential of immuno-therapy for the eradication of CSCs in different tumors.

## 4. Natural Products 

Natural products (NPs) have been a historically rich source of biologically active compounds with diverse chemical structures for the pharmaceutical industry. NPs are also known for their ability to influence multiple signaling pathways simultaneously with negligible side effects. In this review, we will be focusing on the anti-CSC activities of flavonoids and FDA-approved drugs originating from natural sources. Most of the reports reviewed in this study presented data using tissue culture-based systems. Furthermore, several studies of anti-CSC activity of the reviewed NPs were also carried out using animal models. However, only a few examples of clinical studies showing anti-CSC activities of NPs are presented.

Deeper understanding of CSC biology revealed the significance of mitochondrial function in CSC survival, CSC maintenance and in their metabolic and energy demands. Although a number of flavonoids such as quercetin and apigenin induced cell apoptosis in a mitochondrial-dependent manner [42,43], the effect of flavonoids on mitochondria function and their impact on CSC survival and maintenance is not the focus of this review.

## 5. Flavonoids Targeting Cancer Stem Cells (CSC)

Flavonoids are dietary polyphenols present in a wide variety of plants, fruits, vegetables, nuts, and teas [44]. More than 6000 flavonoids were isolated, identified, and divided into subclasses such as flavones, flavonols, flavanols, isoflavones, and isoflavans [45,46]. Quercetin and Kaempferol are the most prominent flavonols in foods, Luteolin and Apigenin are prominent flavones, Genistein and Daidzein are the prominent isoflavones present in soybeans, and catechins are the most prominent constituents of green teas [47]. Flavonoids have been of high scientific interest since the 1990s due to their beneficial effects on human health. Consuming flavonoids may contribute to preventing cardiovascular and neurodegenerative diseases and cancer [48,49]. Recently, there is growing evidence of the preventive effect of flavonoids on cancer stem cells [50,51].

**Quercetin:** Quercetin is a flavonol found in various plant-based foods. Quercetin exhibits anticancer properties both in vivo and in vitro [52], and may exert its anticancer effect through several mechanisms, including suppressing inflammation, inducing apoptosis, acting as an antioxidant, and modulating signaling pathways [53,54]. Cao et al. have studied the effect of quercetin on pancreatic cancer stem-like cells using human pancreatic cancer cell lines. It was found that quercetin inhibited the expression of CSC cell surface markers CD24 and CD133 in pancreatic cancer stem-like cells, and induced pancreatic CSCs differentiation mediated by altered function of β-catenin, a signal transduction pathway which plays an important role in maintenance and progression of pancreatic cancer [55,56].

Erdogan et al. described the effect of quercetin on prostate cancer stem cells (PCSCs) survival and migration. The authors examined the effect of quercetin on CD44+/CD133+ and CD44+ stem cells isolated from prostate cancer cells (PC3 and LNCaP cells, respectively). Quercetin inhibited the survival of PC3 and CD44+/CD133+ in a dose- and time-dependent manner. Midkine (MK) is a multifunctional heparin-binding cytokine with anti-apoptotic, migration-promoting, angiogenic, and other biological functions [57,58]. Administration of quercetin to MK-knockout cells resulted in a higher inhibition of cell proliferation compared with quercetin and MK siRNA alone in both androgen-insensitive and androgen-sensitive cells. The combination of quercetin and MK siRNA could significantly promote apoptosis and inhibit migration of PC3 and CD44+/CD133+ cells via downregulating the expression of PI3K/PTEN, MAPK, and NF-κB signaling pathways [59,60]. Tsai et al. used the prostate cancer stem cells DU145-III isolated from DU145 prostate tumor cell line (DU145-P) to explore the effect of quercetin on prostate cancer stem cells. Quercetin suppressed the migratory and invasive potential and vasculogenic mimicry (VM) in DU145-III cells. It lowered the expression of CSC markers CD44, ABCG2, Sox2, and Nanog, attenuated cancer stem cell-associated spheroid formation, and inhibited the JNK signaling pathway [61]. Wei et al. also examined the inhibition effect of quercetin on the self-renewal of breast cancer stem cells (BCSCs), using mammosphere formation assay in human AS-B145 and ASB244 cells. Quercetin suppressed the size and number of primary and secondary mammospheres in a dose-dependent matter. The same results were obtained using Sca-1+4T1 mouse BCSCs [62].

By using human gastric cancer stem cells (GCSCs) isolated from MGC803, a human gastric cancer cell line, Shen et al. showed that quercetin inhibited GCSC survival by inducing cell mitochondrial-dependent apoptosis through the inhibition of PI3K/Akt signaling [42]. The effect of quercetin on cancer multidrug resistance (MDR) was also evaluated. Li et al. used doxorubicin (Dox)-resistant human breast cancer MCF-7/dox cells to examine the effect of quercetin on MDR reversal and investigate its possible mechanism. In this study, quercetin eliminated BCSC and reversed the MDR in breast cancer cells [63].

In another study, Cao et al. showed that quercetin treatment can overcome pancreatic cancer cell resistance to chemotherapeutic drugs such as gemcitabine [55] by exhibiting significant synergy with the standard chemotherapy drugs and reverse MDR. Moreover, Slusarz et al. identified quercetin as an inhibitor of the hedgehog signaling pathway, implicated in cancer stem biology, in prostate cancer [64].

**Luteolin:** Luteolin (LU) is a flavone found in more than 300 plant species, many of which are available in the human diet [65]. Like quercetin, luteolin was found to suppress CSCs properties and metastasis in the isolated PCSCs, Du145-III [61]. Moreover, luteolin suppressed EMT and cell migration in triple negative cancer cells [66]. Ma et al. reported that luteolin inhibited the survival and self-renewal of liver cancer stem-like cells. Luteolin was also able to affect the number and size of the tumor spheroids [67].

Using human breast cancer xenograft tumors in nude mice, Cook et al. have shown that luteolin reduced breast cancer cell viability, xenograft tumor VEGF expression, and blood vessel density. Furthermore, luteolin blocked MPA-induced acquisition of stem cell-like properties by breast cancer cells. It was also shown that luteolin inhibited various stem cell markers such as CD44, ALDH1, and others in breast cancer cells [68,69,70].

Tu et al. have investigated the effect of luteolin on oral cancer stem cells (OCSCs) using normal human gingival epithelioid S g cells and OCSC cell lines. Luteolin effectively inhibited the proliferation rate, self-renewal, aldehyde dehydrogenase 1 activity, and CD44 positivity of OCSC by the inactivation of IL-6/STAT3 signaling. Luteolin restored radio-sensitivity in OCSC. The combination between luteolin and radiation treatment showed a synergistic effect on invasiveness and clonogenicity of OCSC. Interestingly, luteolin did not cause significant cytotoxicity in normal epithelial cells [71].

Chakrabarti et al. showed that luteolin and silibinin have a synergetic effect on the inhibition of glioblastoma stem cells (GbSC). Their study demonstrated that a combination of luteolin and silibinin effectively inhibited proliferation, migration and invasion and induced apoptosis through downregulation of PKCα and iNOS in human glioblastoma SNB19 cells and GbSC cells [72].

**Apigenin:** Apigenin is a flavone commonly found in plant-derived beverages, some herbs, fruits, and many vegetables, such as parsley, tea, and thyme, and has anticancer properties [73,74].

Erdogan et al. examined the effect of apigenin on prostate CSCs (CD44+) isolated from human prostate cancer cells (PC3). Apigenin inhibited PCSCs and PC3 cell survival and migration in a dose-dependent manner, induced apoptosis via an extrinsic caspase-dependent pathway, and reduced pluripotency marker Oct3/4 protein expression, which may be associated with the downregulation of PI3K/Akt/NF-κB signaling [75].

Ketkaew et al. used the head and neck squamous cell carcinoma cell line HN-30 to examine the effect of apigenin on squamous CSCs. HN-30 cells show expression of stem cell markers induced by hypoxia. Apigenin significantly decreased HN-30 cell viability in a dose- and time-dependent manner and significantly downregulated the expression of CD44, NANOG, and CD105 markers [76].

Kim et al. investigated the effect of apigenin on cancer stem cell-like phenotypes of human glioblastoma (GBM) cell lines U87MG and U373MG. Apigenin inhibited the self-renewal capacity, cell growth, and clonogenicity, and also the invasiveness of GBM stem-like cells. Apigenin inhibited the GBM stem cell-like phenotypes via downregulation of the c-Met signaling pathway. It blocked the phosphorylation of c-Met and its downstream effectors and reduced the expression levels of stem cell markers such as CD133, Nanog, and Sox2 [77].

Apigenin was also reported to sensitize human CD44+ prostate cancer stem cells to cisplatin therapy by inhibition of cancer stem cells of the lung [78]. The combination of apigenin and cisplatin significantly enhanced cisplatin’s cytotoxic and apoptotic effects through downregulation of Bcl-2. The combined therapy suppressed the phosphorylation of PI3K and Akt, inhibited the protein expression of NF-κB, and downregulated the cell cycle by upregulating p21, as well as cyclin-dependent kinases CDK-2, -4, and -6. Apigenin also increased the inhibitory effects of cisplatin on cell migration via downregulation of Snail expression [78,79].

**Other Flavonoids that Affect CSC:** Wogonin is an O-methylated flavone with anticancer properties found in *Scutellaria baicalensis* [80]. It is a well-known drug used for various types of cancers, including hepatic carcinoma, pulmonary carcinoma, and glioblastoma. Wogonin induces apoptosis in the CSCs of human osteosarcoma cells (CD133-positive osteosarcoma cells), inhibits its mobility in vitro via downregulation of MMP-9 expression, and represses its self-renewal ability [81]. Furthermore, an in vitro study revealed that wogonin exhibits an inhibitory function against osteosarcoma stem cells. Wogonin suppressed the expression of stem cell-related genes by regulating reactive oxygen species (ROS) levels and ROS-related signaling [82].

Genistein is a naturally occurring isoflavone, which suppressed tumorsphere formation and decreased Gli1 and CD44 expression [83]. In xenograft models, genistein inhibited tumor growth and downregulated the expression of Gli1 and CD44 in tumor tissues in docetaxel-resistant prostate cancer cells [83] and in a breast cancer model [84]

Myricetin is a flavonol found in berries, onions, and red grapes. Myricetin promotes osteogenic differentiation of human periodontal ligament stem cells via the upregulation of alkaline phosphatase (ALP) activity and expression of osteogenic-related factors through BMP-2/Smad and ERK/JNK/p38 MAPK pathways [85].

Epigallocatechin gallate (EGCG), found in tea leaves, was found to inhibit the self-renewal capacity of human PCSC of the CD44+CD133+ population. EGCG induced apoptosis by activating caspase-3/7 and inhibiting the expression of Bcl-2, survivin, and XIAP in CSCs and inhibited CSC’s migration and invasion. Interestingly, EGCG synergizes with quercetin in eliminating cancer stem cell-characteristics [86,87]. Moreover, EGCG was found to downregulate the expression of Gli1 and inhibit the proliferation of a number of cancer cell types [64,88]. EGCG inhibited cellular self-renewal capacity through regulating stem cell markers, Nanog, c-Myc and Oct4, as well as Hh signaling mediators, Smo, Ptch and Gli1/2 [89]. In an animal model of carcinogen-induced liver cancer, EGCG reduced the population of CD44-positive cells and inhibited the expression of Gli1, Smo, cyclin D1, cMyc, and EGFR [90].

Fisetin is a flavonol that is naturally abundant in many fruits and vegetables and has anti-tumor properties [91,92]. Si et al. examined its effect on human renal CSC (HuRCSC) isolated from renal cancer samples and found that fisetin inhibited the proliferation by an epigenetic mechanism. It significantly decreased the expression of ten-eleven translocation protein1 (TET1), effectively inhibited the 5−hydroxymethylcytosine (5hmC) modification levels at the CpG islands in cyclin Y (CCNY) and CDK16, and reduced their transcription and activity, which caused a cell cycle arrest [93].

Tabasum et al. explored fisetin anti-metastatic effects in non-small cell lung carcinoma (NSCLC) cell lines A549 and H1299 with emphasis on epithelial to mesenchymal transition (EMT). EMT promotes metastasis by allowing the tumor cells to acquire increased migratory and invasive properties, mediating their dissemination to faraway sites [94,95]. It was found that fisetin significantly inhibited the migration and invasion of NSCLC cells under non-cytotoxic concentrations. Fisetin attenuated EMT in both cell lines with upregulated expression of epithelial markers and downregulation of mesenchymal markers. Furthermore, fisetin treatment downregulated NSCLC stem cell signature markers CD44 and CD133. Thus, fisetin is a potential therapeutic agent for lung cancer stem cells [96].

Broussoflavonol B from the bark of the Paper Mulberry tree (*broussonetia papyrifera*), exhibited potent growth inhibitory activity towards breast cancer cells, sensitized breast cancer stem/progenitor cells to tamoxifen, and restricted the proliferation of ER-negative breast cancer stem-like cells [97].

Morusin, a butenylated flavonoid isolated from the root bark of Moraceae, also has the potential to target cervical CSCs by attenuating NF-kB activity [98].

Icaritin a prenylflavonoid derivative from Epimedium Genus, inhibited growth of hepatic cancer stem cells through downregulating STAT activation [99] and inhibited malignant growth of hepatocellular carcinoma-initiating cells (HCICs) [100]. Its analogue SNG1153 inhibited tumorsphere formation and decreased CD133-positive (lung CSC marker) cancer cells. It also inhibited the growth of lung CSCs, which may be a novel therapeutic agent to treat human lung cancer. SNG1153 induced β-catenin phosphorylation and downregulated β-catenin [101].

Casticin, which is derived from *Fructus Viticis Simplicifoliae*, inhibited the self-renewal of liver cancer stem cells from the MHCC97 cell line, and β-catenin was identified as the potential target [102].

## 6. Flavonoids Targeting ABCG2 in CSCs

ABC transporters consist of 49 transporter proteins that are classified into seven subfamilies, ABCA to ABCG, that locate in the cell membrane and have diverse functions [103]. By using ATP, ABC transporters work to transport their substrates across the cell membrane and to protect cells against xenobiotics, including some anti-cancer drugs [104].

Cancer stem cells are known to express elevated levels of ABCG2 and consequently are characterized with multi-drug chemoresistance [105]. These cells are thought to lead to a relapse after chemotherapy. Therefore, inhibition of ABCG2 could have an additional benefit besides counteracting multidrug resistance, selective killing of CSC. Interestingly, an increasing number of FDA-approved tyrosine kinase inhibitors (TKIs), including imatinib and gefitinib, reported to downregulate or inactivate ABCG2 [106] and, therefore, may serve as candidates to reverse cancer stem cell chemoresistance. Similarly, a number of natural products were also reported to inactivate ABCG2 and thus sensitize cancer stem cells to activity of standard chemotherapy including estrogenic compounds; several tamoxifen derivatives in addition to phytoestrogens and flavonoids have been shown to reverse ABCG2-mediated drug resistance. Flavonoids seem promising ABCG2 inhibitors, as they exhibit selective and broad-spectrum activity [107].

The flavonoids silymarin, hesperetin, quercetin, and daidzein were shown to increase the intracellular accumulation of mitoxantrone in ABCG2-expressing cells [108]. Flavonoids such as Chrysin and biochanin A reported as potent inhibitors of ABCG2 in breast cancer cells, and were consequently able to sensitize breast cancer stem cells to cancer chemotherapy activity such as mitoxantrone [109]. Interestingly, inhibitory flavonoids appear either non-competitive or partially competitive towards mitoxantrone efflux. Most compounds do not inhibit ATPase activity in ABCG2, and are assumed not to be transported themselves by the transporter.

Structure activity studies led to the identification of novel ABCG2 inhibitors such as 6-prenylchrysin [110] exhibiting an IC_50_ of 0.3 M. The relatively low toxicity of 6-prenylchrysin and efficient sensitization of cell growth to mitoxantrone made these compounds promising for future potential use in clinical trials.

In a recent study, the inhibitory effect of naturally occurring flavonoids on ABCG2 was correlated with their positive effects on the pharmacokinetics of anticancer drugs [111]. A panel of 32 flavonoids was screened by using topotecan accumulation and cytotoxicity assays, and led to the identification of 3′,4′,7-trimethoxyflavone as the most potent inhibitors of ABCG2.

It was found that multiple flavonoid combinations induce strong ABCG2 inhibition by increasing both accumulation and cytotoxicity of mitoxanthrone in ABCG2-overexpressing breast cancer cells. The best candidates were biochanin A (isoflavone), kaempferide (flavonol), 5,7-dimethoxyflavone and 8-methylflavone [112].

Chalcones, which also belong to the flavonoids family and are natural compounds present in edible plants, were found to inhibit differentially ABCB1 and ABCG2, basic chalcones being more efficient on ABCB1 transporter [113] and non-basic chalcones on the ABCG2 transporters [114]. Chalcones exhibiting the highest activity and selectivity toward ABCG2 were found among derivatives which are dimethoxylated or dihydroxylated at the A-ring, as evidenced by the mitoxantrone accumulation and cytotoxicity assays.

## 7. Anti-CSC Activity of Flavonoids Mediated by Modulation of microRNAs

MicroRNAs (miRNAs) are short (~17–28 nucleotides) endogenous RNA molecules which regulate mRNA stability and translation as part of the RISC complex, by binding to specific sites in the 3′ untranslated regions of the mRNAs through partial sequence complementarity [115]. Since the discovery of miRNAs at the turn of the century, their involvement in a variety of biological processes has been described, and it is estimated that the expression of >60% of all protein-coding genes is regulated by miRNAs [116], and depending on the inclusion criteria, between 600 and 2000 miRNAs are encoded in the human genome [117,118].

One of the better studied contexts of miRNA function has been the biology of cancer. A panel of miRNAs has been found to be upregulated in various types of cancer, with their higher levels contributing to the different aspects of oncogenesis (thus, they were dubbed “oncomiRs”) [119]. Other miRNAs have been identified as downregulated in cancer and function as tumor suppressors [120]. Finally, some miRNAs (such as miR-10b and miR-221/222) can function as both tumor suppressors and oncomiRs, based on the cancer type and stage, and depending on the specific selection and repertoire of expressed target genes [121,122,123,124].

A number of miRNAs have been shown to function in the maintenance of cancer stem cells (CSC) [125]. Thus, the miR-34, miR-199, and miR-200 families, as well as miR-1, miR-143 and miR-146 have all been shown to regulate elements of the Notch pathway, which plays a central role in CSC [126]. Other miRNAs have been implicated in the regulation of EMT, among them miR-106b, miR-22, and miR-203. miR-9/9* and miR-21 have been reported to promote the CSC phenotype, whereas miR-10b, miR-328 and miR-495 have been shown to promote metastasis and drug resistance of the cells. The Let-7 family and miR-302 have been reported as involved in CSC differentiation (review, [127]). Additionally, several tumor suppressors (e.g., let-7, miR-34, miR-146, miR-200) and oncomiRs (e.g., miR-21, miR-210) are involved in reactive oxygen species signaling, which may induce CSC properties and EMT (review, [128,129]).

Several known tumor suppressor miRNAs are direct regulators of the expression of CSC marker proteins. Thus, based on the TargetScan prediction algorithm [130], the PROM1 gene encoding the CD133 peptide has a conserved miR-200 binding site in its 3′UTR, and the targeting was validated in a rat hepatic oval cell model [131]. Similarly, CD44 was reported to be directly regulated by miR-221/222 in bone marrow cells [132].

**Effects of Flavonoids on Micrornas:** Several studies have examined the effects of flavonoid exposure on the levels of specific miRNAs and their downstream targets. Thus, the Koike group has reported that treatment with the flavonoid apigenin suppresses the exogenous overexpression of miR-103 in mice, resulting in improved glucose tolerance [127]. The same group has also reported that apigenin treatment suppresses miR-122 levels in vitro, likely via a mechanism involving TRBP phosphorylation [129]. Although that particular study was focused on the role of miR-122 in HCV infection, a tumor suppressor role has previously been reported for this miRNA [133,134].

The ability of flavonoids to affect miRNAs involved in carcinogenesis (review, [135,136]) is particularly relevant in CSCs. Thus, in breast cancer cells, exposure to Glabridin, a phytochemical from the root of Glycyrrhiza glabra, upregulated miR-148a via promoter de-methylation, leading to the suppression of SMAD2 and decrease in CSC-like properties [137]. This attenuation was observed both in vitro, i.e., in MDA-MB-231 and Hs-578T breast cancer cell lines, and in mouse xenograft models. The upregulation of miR-148 in breast cancer cell lines by Glabridin also suppressed the Wnt/β-catenin signaling pathway, resulting in decreased angiogenesis [138].

In non-small-cell lung cancer, EGCG, enhanced the levels of mir-485-5p, suppressing the levels of two oncogenic targets, CD44 and the nuclear receptor RXRα, both effects contributing to the decrease in CSC-like properties [139,140]. Inhibitors of miR-485 increased CSC-like phenotypes, which could be reversed by indicated doses of EGCG [140]; this link was also tested in vivo. Finally, in pancreatic duct carcinoma cell lines, a combination of sulforaphane, quercetin and catechin treatments (including EGCG) led to the upregulation of the miRNA let-7 and a decrease in the CSC-like self-renewal properties [141]. Quercetin also induced miR-200b-3p, decreasing Notch signaling, promoting daughter cell asymmetry and inhibiting the self-renewal in pancreatic cancer cells [142].

In CD133+ melanoma cells, Morin treatment induced miR-216a. When carried out in vitro, this led to a reduction in cell viability, in sphere formation, and in the expression of stem cell marker genes CD20, CD44, CD133 and Wnt-3A. This was also observed in vivo: a melanoma xenograft model treated by Morin showed reduced tumor size and weight, as well as reduced expression of stem cell markers and Wnt-3A [143].

Non-toxic natural compounds have shown promise as a supplementary approach to conventional chemotherapy, in suppression of CSCs by means of targeting miRNAs. Thus, treatment with resveratrol or its analogue pterostilbene elevated the expression and activity of Argonaute2, increasing the expression of a number of tumor-suppressive miRNAs, e.g., miR-16, -141, -143, and -200c, and resulting in long-term suppression of their targets in breast cancer [144]. The most relevant findings are summarized in Table 2, and a graphical summary appears in Figure 2.

Given the ubiquitous participation of miRNAs in the signaling pathways affecting CSCs, it is reasonable to assume that the effects of flavonoids on CSCs are mediated, inter alia, by miRNAs, and that these miRNAs are potential targets for supplementary therapeutic intervention.

## 8. FDA-Approved Drugs Based on Natural Products Modulating CSCs

The current manuscript focuses on anti-CSCs activity of flavonoids. However, no flavonoids with anti-CSC activity have been approved by the FDA. Thus, we will discuss plant-based FDA-approved drugs that exhibit anti-CSC activity. Among them, we chose to focus on Chloroquine (CQ) and metformin.

**Chloroquine (CQ):** CQ was isolated from the bark of the Cinchona tree (cinchona Officinalis) and is an FDA-approved drug for the treatment and prevention of malaria. CQ is a member of the 4-aminoquinoline drug class. Currently, it is also being studied as a treatment against COVID-19 [145]. CQ functions as an autophagy inhibitor. Recent data indicated that chemotherapy resistance in epithelial ovarian cancer (EOC) is associated with autophagy activation. CSC isolated from EOC ascites exhibit higher basal autophagy compared with the non-stem counterpart. Inhibition of this pathway, by ex vivo CQ treatment or CRISPR/Cas9 ATG5 knockout, impaired CSC viability and the ability to form tumorspheres in vitro, and significantly reduced tumorigenic potential in vivo. Also, autophagy inhibition showed a synergistic effect with carboplatin administration on both in vitro CSC properties and in vivo tumorigenic activity [146]. CQ was also effective in inducing mitochondrial structural damage and autophagy inhibition in CSC from triple-negative BC (TNBC). CQ effectively diminishes the TNBC cells’ ability to metastasize in vitro and in a TNBC xenograft model. Thus, as TNBC is highly enriched with CSCs, CQ may be an effective addition to carboplatin for effective treatment [147]. Additional reports showed that CQ sensitizes CSC TNBC cells to paclitaxel through inhibition of autophagy and reduces the CD44(+)/CD24(-) CSC population in both preclinical and clinical settings [148]. Several molecular mechanisms were implicated in mediating CQ function. Choi et al. reported that CQ regulates the CSCs in TNBC through inhibition of the Janus-activated kinase 2 (Jak2) signal transducer and activator of transcription 3 (STAT3) signaling pathway by reducing the expression of Jak2 and DNA methyltransferase 1 (DNMT1) [148]. Other reports demonstrated that CQ exerts anti-CSC activity to CSC pancreatic cancer by inhibiting the CXCL12/CXCR4 signaling, resulting in reduced phosphorylation of ERK and STAT3 [149]. Furthermore, CQ showed potent inhibition of hedgehog signaling by decreasing the production of Smoothened, translating into a significant reduction in sonic hedgehog-induced chemotaxis and downregulation of downstream targets in CSCs and the surrounding stroma [149]. Thus, CQ is an effective adjuvant therapy to chemotherapy, exhibiting anti-CSC activity and the clinical significance of this drug should be further explored in the clinical setting.

**Metformin:** metformin (MET, N′-dimethyl biguanide) was first synthesized in 1920s from guanidine derived from French lilac (*Galega officinalis* L.) and is one of the most commonly used oral antihyperglycaemic drugs used in the treatment of type 2 diabetes [150], particularly in people who are overweight [151]. Metformin inhibits gluconeogenesis, reduces circulating levels of glucose, increases insulin sensitivity, and reduces hyperinsulinemia [152]. The mechanisms of metformin action involve AMP-activated protein kinase (AMPK)-dependent and AMPK-independent signaling pathways [153].

Metformin was reported to exert anti-cancer activity, in particular against cancer stem cells (CSCs) including breast cancer (BC) stem cells. Metformin suppresses self-renewal and stemness, and reduces the percentage of cells in the S phase [154]. BC CSCs are particularly sensitive to metformin, which induces rapid cell death facilitated through several pathways including Notch 1, NF-κB, Sox2, KLF-4, Oct4, Lin28, MMP-9, and MMP-2 [155]. Moreover, metformin mediates its activity against CSCs through regulation of many miRNAs, including let-7 and miR-193b [155,156]. Interestingly, among the molecular BC subtypes, TNBC shows the highest enrichment of CSCs [157] which show elevated sensitivity to metformin.

In combination with chemotherapy, metformin is especially active against BC CSCs [158]. In studies of trastuzumab-resistant BC cells as well as xenograft models, the combination of trastuzumab and metformin significantly reduced CSC subpopulations and reduced tumor volume [158,159,160]. In combination with doxorubicin, paclitaxel, or carboplatin, metformin can also eradicate CSCs and reduce the effective dosage required of the chemotherapeutic agents, minimizing potential organ toxicity [158,159].

Epidemiologic data show a significant lowering of cancer risk in patients with metabolic dysregulation who take metformin [161,162,163,164]. Metformin use by BC patients has also been associated with improved treatment response and survival. In one meta-analytic study of BC patients with diabetes, metformin use was associated with a 65% improvement in BC-specific survival compared with nonusers [165]. The anticancer properties of metformin are in contrast to other antidiabetic agents, including sulfonylureas and insulin, which promote cancer growth [166].

The anti-cancer activity of metformin was explored in the clinics as an adjuvant therapeutic option for the management of BC [167,168,169]. Clinical trials have shown that metformin improves outcomes for BC patients with diabetes [170]. Notably, TNBC, which is typically the most aggressive and is less responsive to traditional chemotherapy, is particularly sensitive to metformin.

Clinical activity of metformin was not limited to diabetic BC patients, as nondiabetic BC patients were also evaluated with promising outcomes. A meta-analysis of studies involving metformin therapy in nondiabetic patients and diabetic BC patients, is presented in [167]. Numerous clinical trials are currently underway in BC patients and other types of cancer to evaluate the benefit of metformin combined with or following the administration of other therapeutic agents [161,167,171,172,173,174].

## 9. Conclusions and Future Perspective

The existence of cancer stem cells and their involvement in chemoresistance and recurrence of the disease is well documented. This understanding should lead to changes in the strategy of cancer therapy, to include the means to eradicate cancer stem cells in order to achieve a lasting response. This may be achieved by the development of effective anti-CSCs therapies that will be combined with existing anti-cancer therapies. Natural flavonoids or potent derivatives are good candidates in exhibiting anti-CSC activity and targeting key functions required for CSC survival (Figure 3). Despite the promising potential of flavonoids, it is a known fact that numbers of drawbacks limit the full therapeutic utilization of flavonoids such as lack of selectivity, poor potency, and bioavailability. Overcoming the above limitations might be possible by synthesizing derivatives with improved characteristics. Avenues to improve the bioaccessibility and bioavailability of flavonoids might include increasing the intestinal absorption [175], improving metabolic stability [176], or changing the site of absorption (from colon to small intestine) [177]. To achieve these goals microencapsulation, emulsion/nanoemulsion, solid lipid nanoparticles, liposome/nanoliposome delivery systems, enzymatic methylation, and other consumer-friendly technologies are necessary [178]. Future developments may include the development of bi-modular (hybrid) molecules, resulting in more efficient and lasting therapy. Another challenge is predicting harmful drug–flavonoid interactions. In addition, the increased risk of adverse effects from the use of putative anti-CSC inhibitors in combination with other cancer therapeutics needs to be evaluated.

## Figures and Tables

**Figure 1 ijms-22-13044-f001:**
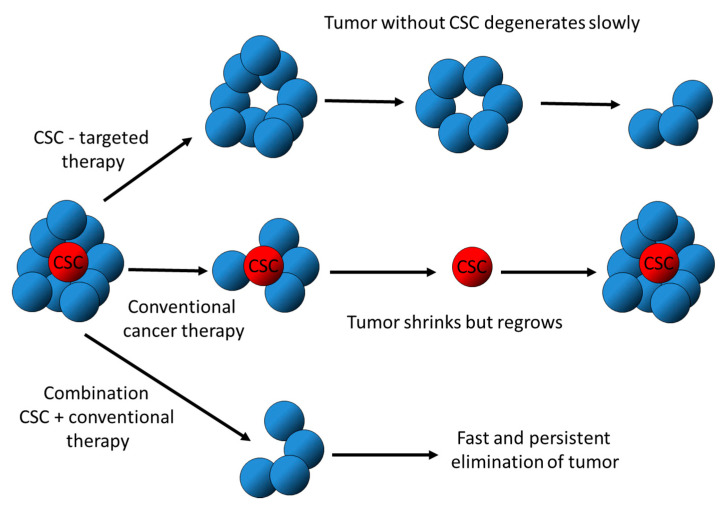
Natural products targeting CSC to augment conventional cancer therapeutics.

**Figure 2 ijms-22-13044-f002:**
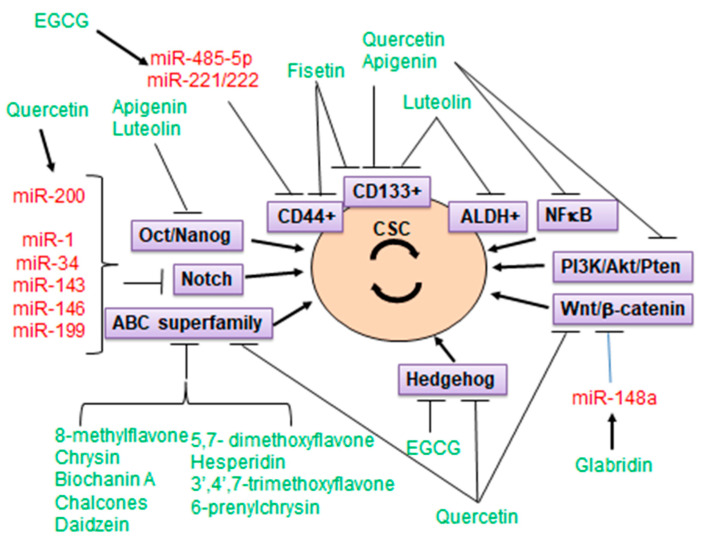
Graphical summary of flavonoids targeting major pathways in CSCs, including those regulated by microRNAs. Major pathways and CSC markers are in purple boxes; flavonoid names are in green; microRNAs are in red.

**Figure 3 ijms-22-13044-f003:**
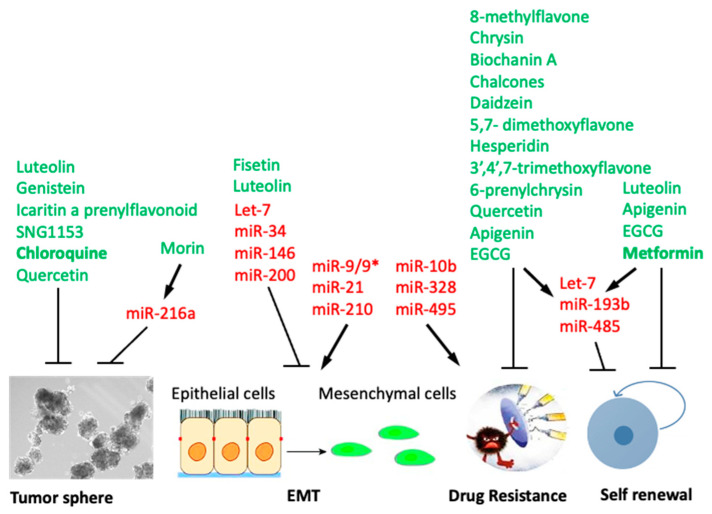
Graphical summary of flavonoids targeting major hallmarks of the CSC phenotype, including those regulated by microRNAs. Flavonoid names are in green; microRNAs are in red.

**Table 2 ijms-22-13044-t002:** A summary of findings regarding NP effects in CSC, including those mediated by microRNAs.

Name	Structure	Activity (Reference)
Quercetin	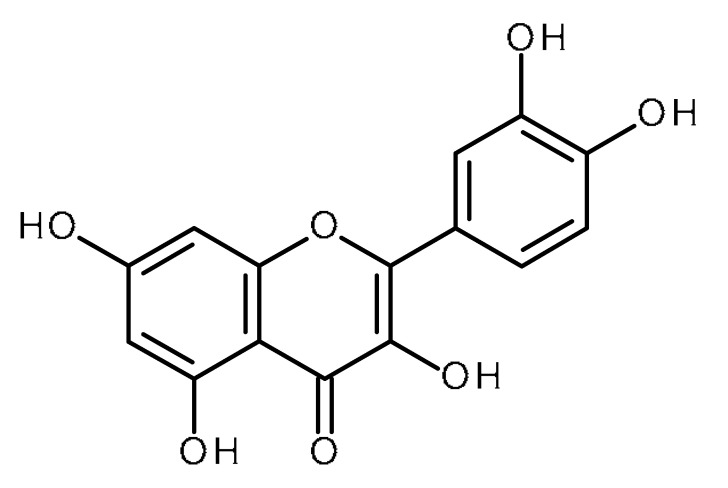	Pancreatic cancer stem-like cells [30] Prostate cancer stem cells (PC3 and LNCaP cells) [31,32] Prostate cancer stem cells (DU145-III cells) [34] Breast cancer stem cells [35] Human gastric cancer stem cells [36] Regulates Let-7, miR-200b-3p in pancreatic duct carcinoma [141,142]
Luteolin	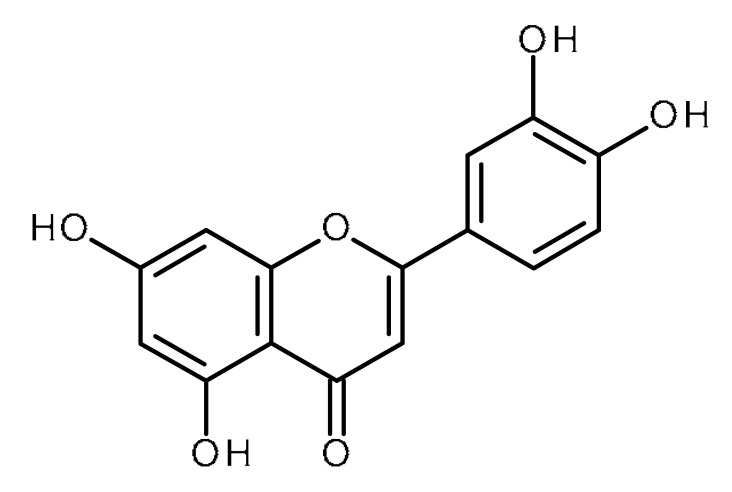	Prostate cancer stem cells (DU145-III cells) [34] Liver cancer stem-like cells [39] Breast cancer stem-like cells [40,41] Oral cancer stem cells [42] Glioblastoma cancer stem cells [43]
Apigenin	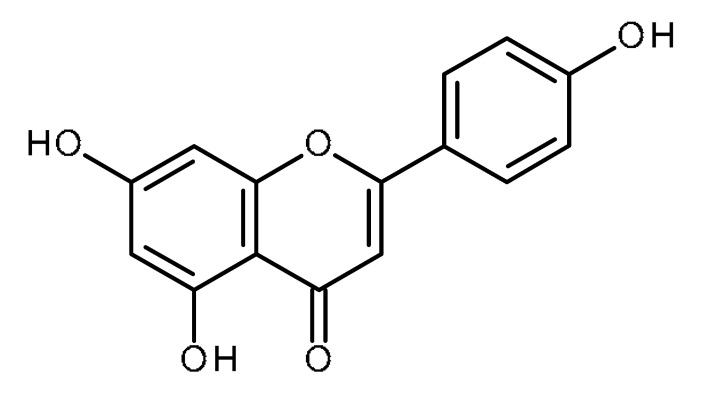	Prostate cancer stem cells (PC3 cells) [46] Squamous cancer stem cells [47] Glioblastoma cancer stem cells [48] Prostate cancer stem cells [49] Suppresses the exogenous overexpression of miR-103 in mice [80] Suppresses miR-122 levels in vitro [82]
Wogonin	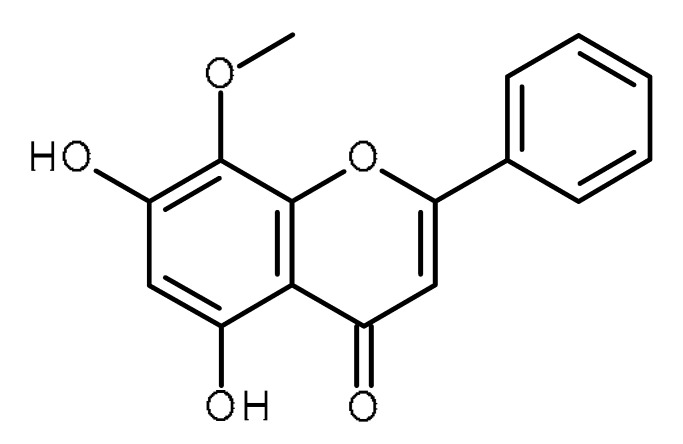	Human osteosarcoma cancer stem cells [51]
Myricetin	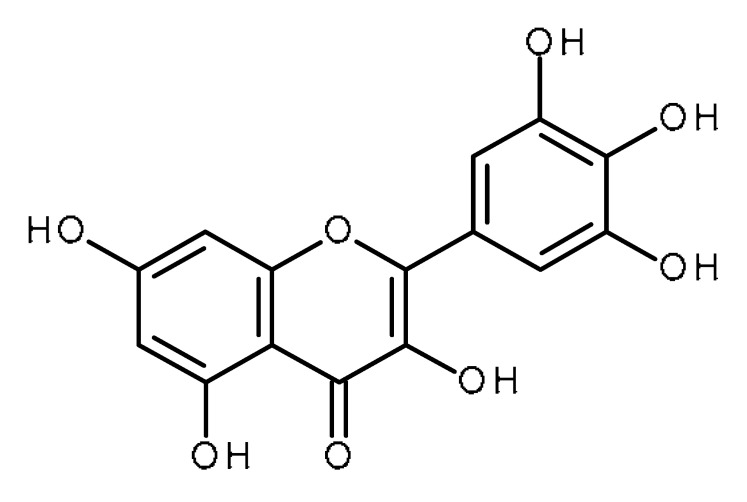	Human periodontal ligament stem cells [52]
Fisetin	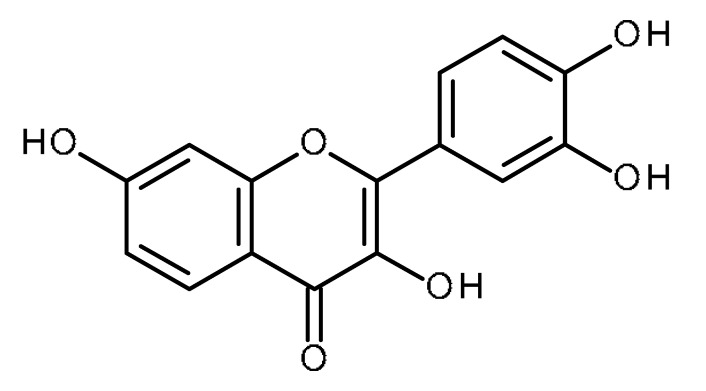	Human renal cancer stem cells [55] Non-small cell lung carcinoma cells (A549 and H1299) [56,57,58]
Epigallocatechin gallate (EGCG)	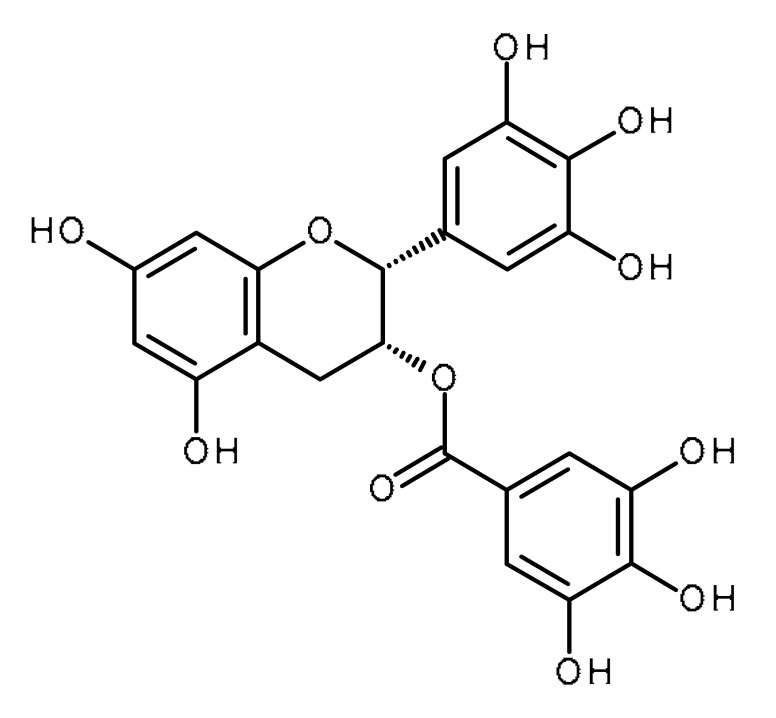	Prostate cancer stem cells [53] Regulates mir-485-5p, Let-7 in non-small-cell lung cancer, pancreatic duct carcinoma [139,140]
Biochanin A	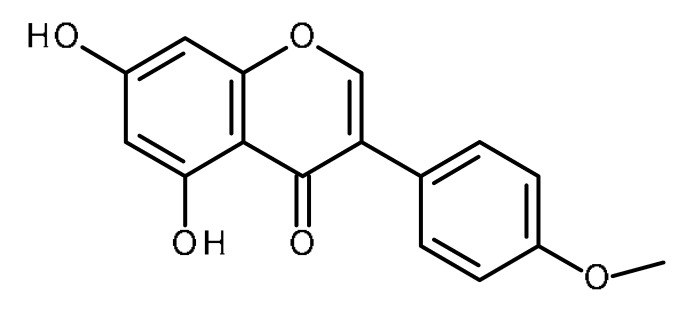	Inhibition of ABCG2 in breast cancer cells [62,65]
Kaempferide	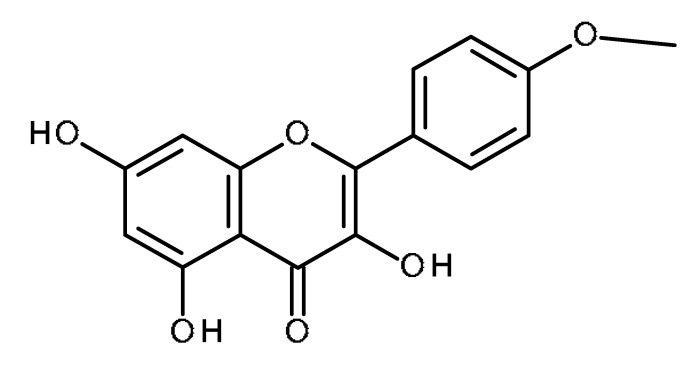	Inhibition of ABCG2 in breast cancer cells [65]
5,7- dimethoxyflavone	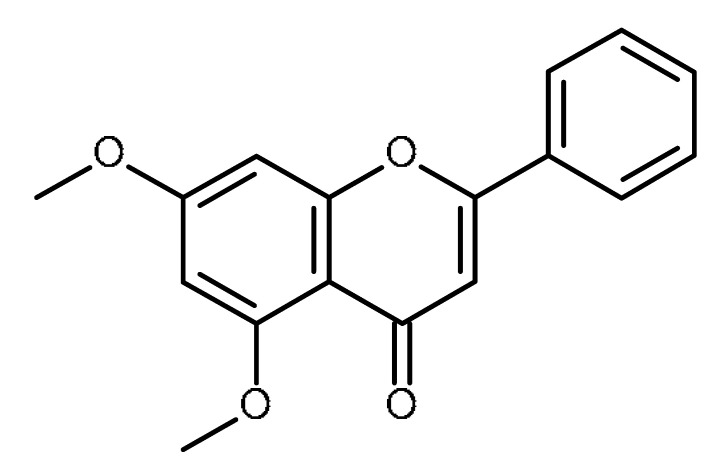	Inhibition of ABCG2 in breast cancer cells [65]
8-methylflavone	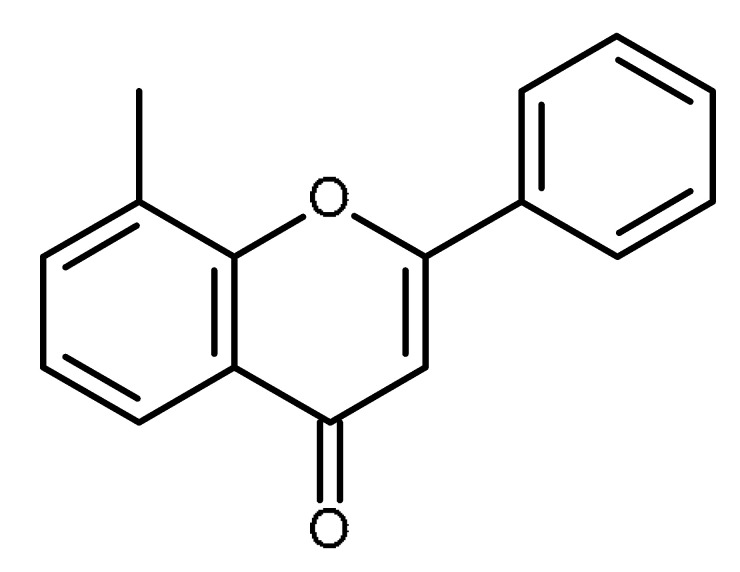	Inhibition of ABCG2 in breast cancer cells [65]
Silymarin (Silybin)	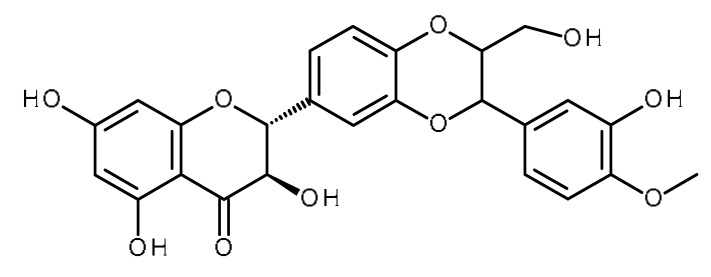	Increase the intracellular accumulation of mitoxantrone in ABCG2- expressing cells [1]
Hesperetin	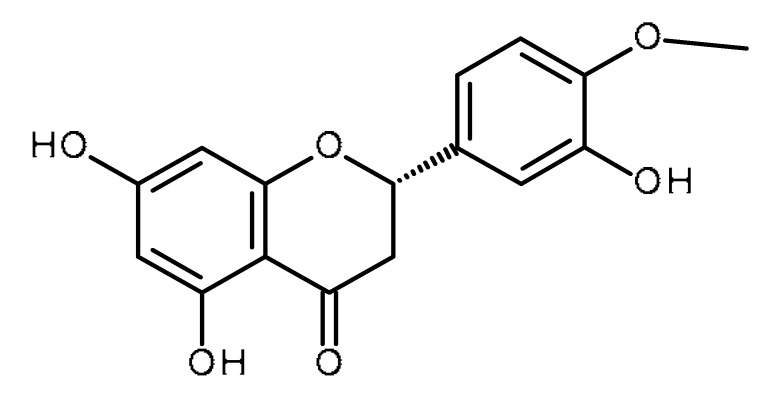	Increase the intracellular accumulation of mitoxantrone in ABCG2- expressing cells [1]
Daidzein	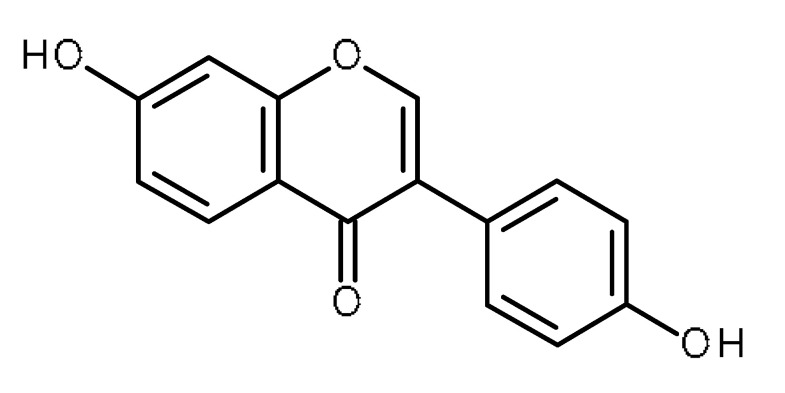	Increase the intracellular accumulation of mitoxantrone in ABCG2- expressing cells [1]
Chrysin	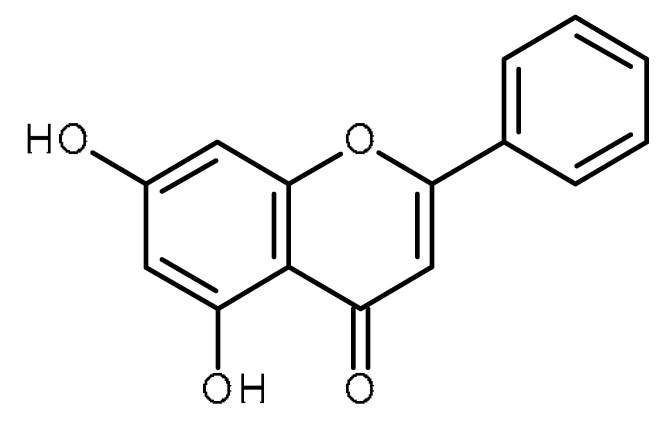	Inhibitor of ABCG2 in breast cancer cells [62]
6-prenylchrysin	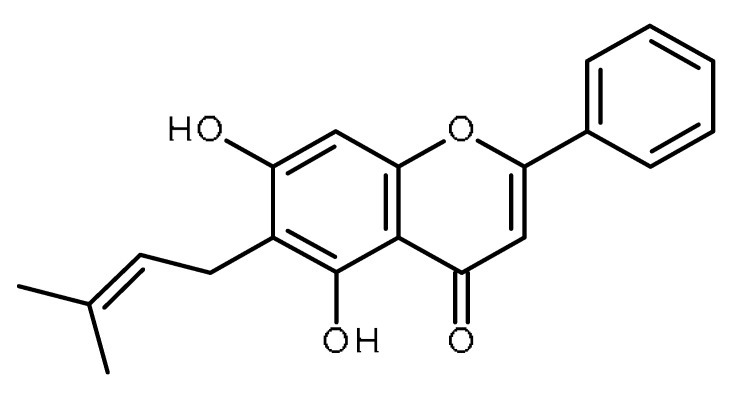	Inhibitor of ABCG2 [63]
3′,4′,7-trimethoxyflavone	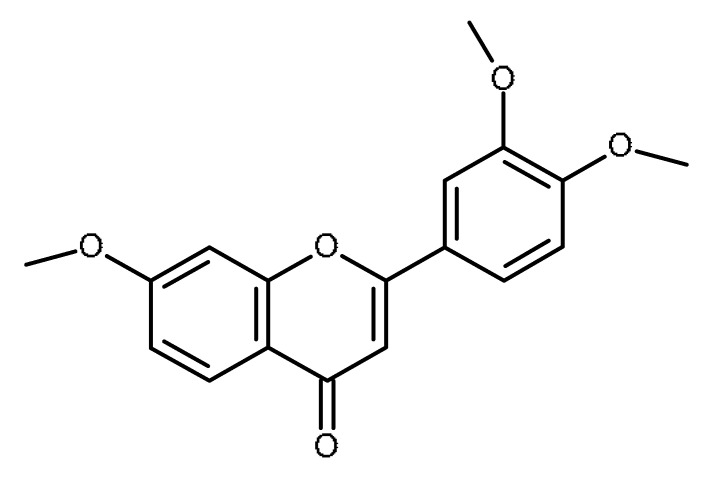	Inhibitor of ABCG2 [64,65]
Icaritin	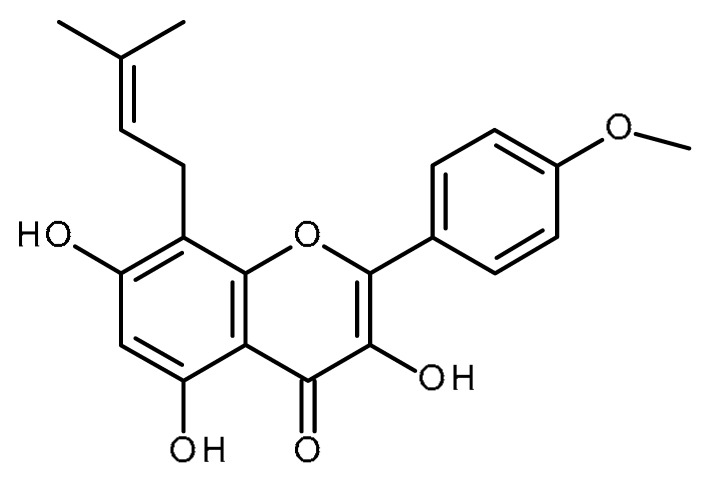	Inhibitor of hepatic cancer stem cells [97,98]
SNG1153	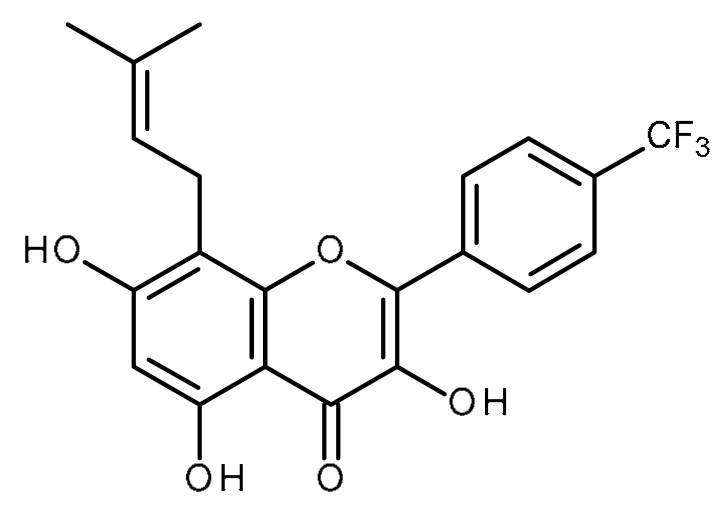	Inhibitor of lung CSCs [99]
Morusin	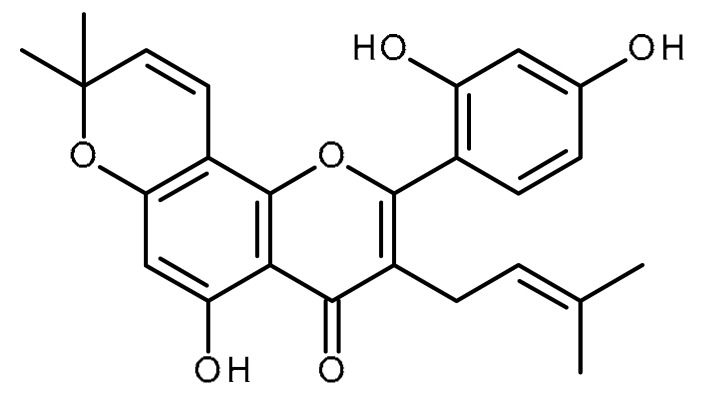	Inhibitor of CSCs by attenuating NF-kB activity [96]
Casticin	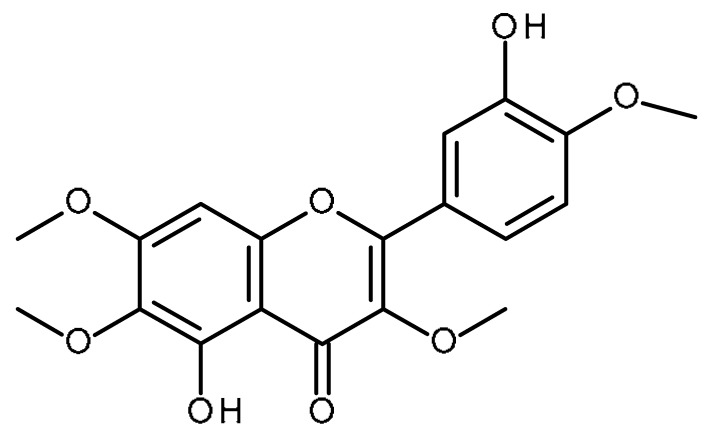	Inhibitor of liver cancer stem cells [100]
Chalcones	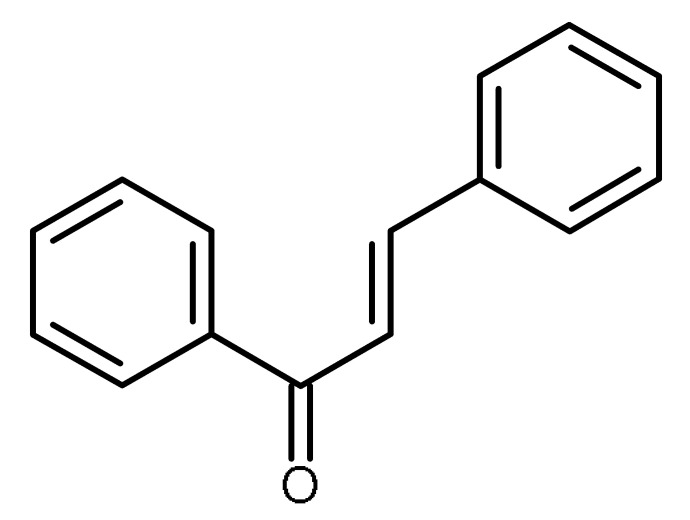	ABCG2 inhibitors [66,67]
Chloroquine (CQ)	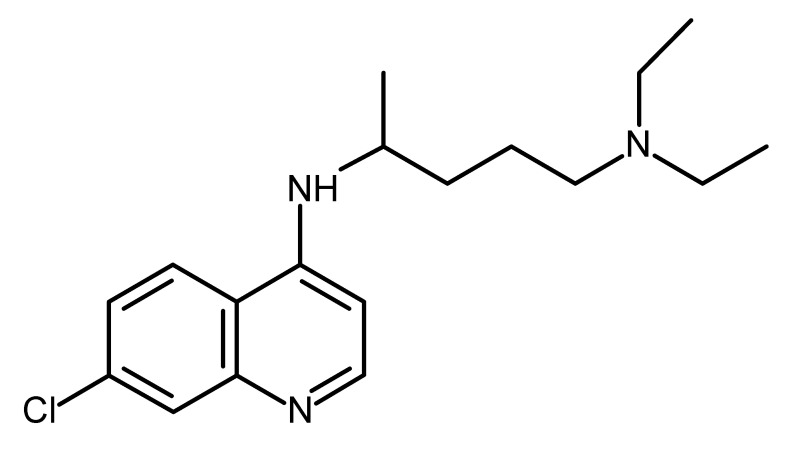	FDA Approved drugs originated from natural products modulating cancer stem cells [97,98,99,100,101]
Metformin	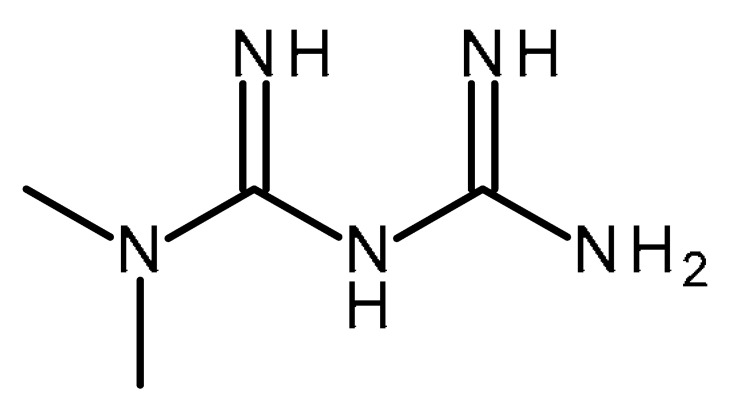	FDA Approved drugs originated from natural products modulating cancer stem cells [102,103,104,105,106,107,108,109,110,111,112,113,114,115,116,117,118,119,120,121,122,123,124]
Glabridin	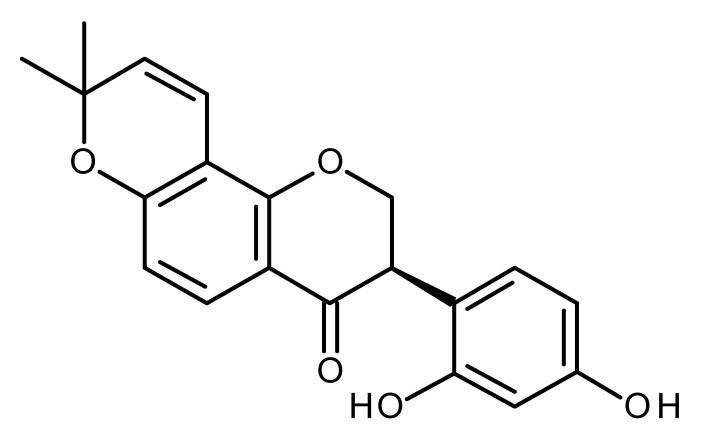	Regulates miR-148 in breast cancer [137,138]
Sulforaphane	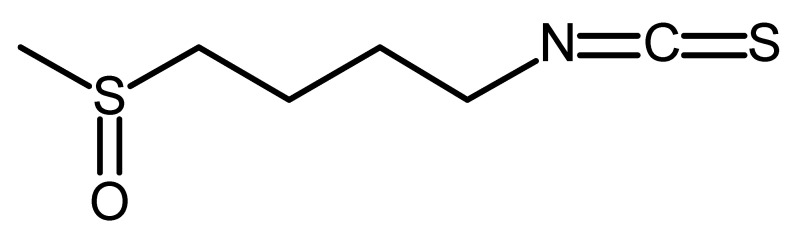	Regulates Let-7 in pancreatic duct carcinoma [141]
Morin	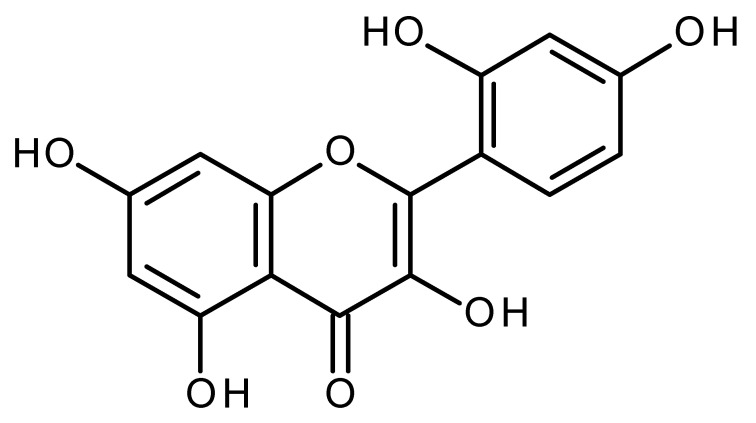	Regulates miR-216a in melanoma [143]
Resveratrol (R=H)/Pterostilbene (R=CH_3_)	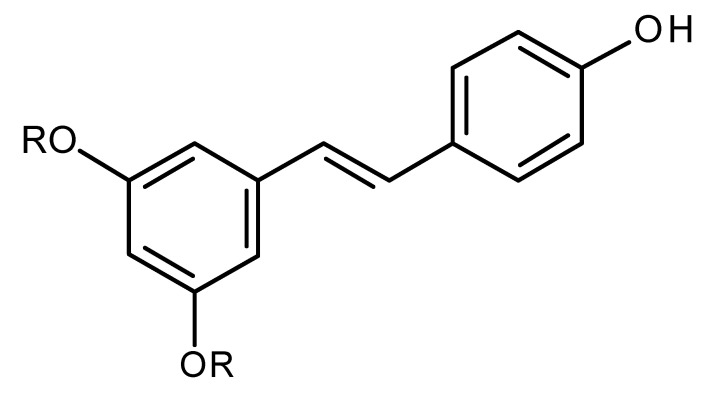	Regulates miR-16, miR-141, miR-143, miR -200c in breast cancer [144]

## Data Availability

Not applicable.

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
