# Peer review of "Natural Products Targeting Cancer Stem Cells for Augmenting Cancer Therapeutics"

_ijms, 2021, doi:10.3390/ijms222313044_

Round 1

Reviewer 1 Report

This manuscript is well-written, but would benefit from a few improvements:

  • Please split the introduction into two separate parts: 1) Hallmark of CSC and 2) Markers of CSC (which would include CD44, CD133, CD24 (currently absent), EPCAM, ALDH, and side population)
  • This manuscript would benefit from more recent references (only 21 are 2018 or newer).

Author Response

We thank the reviewers for their constructive comments. We have now thoroughly revised the manuscript in order to address the issues that were raised. A point-by-point response follows.

Reviewer 1:

Comments and Suggestions for Authors

This manuscript is well-written, but would benefit from a few improvements:

  • Please split the introduction into two separate parts: 1) Hallmark of CSC and 2) Markers of CSC (which would include CD44, CD133, CD24 (currently absent), EPCAM, ALDH, and side population)

This was done as per the reviewer’s request.

  • This manuscript would benefit from more recent references (only 21 are 2018 or newer).

We thank the reviewer for averting our attention. Additional updated references are now included.

Reviewer 2 Report

I have read with interest the review entitled “Natural products targeting cancer stem cells for augmenting cancer therapeutics”, and I have several comments to it:

  1. There are no Figures in the review, making the review text-only. I believe it is almost mandatory to add at least some figures. I believe authors should graphically summarize the critical properties of CSCs and the natural products that affect them. I think the mechanisms by which these natural compound interfere with CSCs should be described a bit more in detail. The role of mitochondria and metabolic rewiring in the effect of the compounds should be also discussed and possibly graphically shown as a Figure. Another important Figure should be the signaling pathways that are targeted by these flavonoids or other natural compounds and the place where there compounds act should be shown. I believe without graphical input in the article, it is hard to envision that readers would be interested to read it..
  2. From the scientific point of view the topic is interesting, yet there might be a significant drawback in the availability of the individual flavonoids in the body, as they are usually not well soluble and the average concentrations that could be achieved systemically are too low to bring about the effect expected based on the in vitro This should be commented for each compound and should be deeply discussed.
  3. Are there any preferences why some compounds have not been included in the review? For example curcumin, phenolic compounds, ascorbic acid and limonoids are not discussed at all?? Are data on the effects of these compounds on CSCs lacking even in the literature? I believe other compounds with significant effect should be discussed!

Author Response

Reviewer 2:

Comments and Suggestions for Authors

I have read with interest the review entitled “Natural products targeting cancer stem cells for augmenting cancer therapeutics”, and I have several comments to it:

  1. There are no Figures in the review, making the review text-only. I believe it is almost mandatory to add at least some figures. I believe authors should graphically summarize the critical properties of CSCs and the natural products that affect them. I think the mechanisms by which these natural compound interfere with CSCs should be described a bit more in detail. The role of mitochondria and metabolic rewiring in the effect of the compounds should be also discussed and possibly graphically shown as a Figure. Another important Figure should be the signaling pathways that are targeted by these flavonoids or other natural compounds and the place where there compounds act should be shown. I believe without graphical input in the article, it is hard to envision that readers would be interested to read it..

We thank the reviewer for this excellent recommendation. We have incorporated two figures in the revised manuscript: Figure 1 presenting the potential benefit of a combination therapy, and Figure 2 which was built following the detailed suggestions of the reviewer.

  1. From the scientific point of view the topic is interesting, yet there might be a significant drawback in the availability of the individual flavonoids in the body, as they are usually not well soluble and the average concentrations that could be achieved systemically are too low to bring about the effect expected based on the in vitro This should be commented for each compound and should be deeply discussed.

The important issue of bioavailability is now discussed in the Discussion section.

  1. Are there any preferences why some compounds have not been included in the review? For example curcumin, phenolic compounds, ascorbic acid and limonoids are not discussed at all?? Are data on the effects of these compounds on CSCs lacking even in the literature? I believe other compounds with significant effects should be discussed!

We thank the reviewer for this comment. Including all types of natural products and all their mechanisms of action was not the original intention of this mini-review. A statement which more accurately clarifies the scope of the review and its limits, now appears in the Natural Products section of the manuscript.

Reviewer 3 Report

Meerson et al. Natural products targeting cancer stem cells for augmenting cancer therapeutics

This review described the hallmark of cancer stem cells (CSC) and discussed the potential of natural products especially flavonoids, chloroquine and metformin in targeting multiple signaling pathways regulating cancer stem cell self-renewal, proliferation, survival, and their potential to augment the efficacy of standard cancer therapy. The review is timely, nicely written and very comprehensive.

Major comments:

  1. Though the authors discussed CD44 and CD133, ESA and ALDH, other CSC biomarkers including CD24, CD49, and CD90 etc. as well as biomarkers for tissue specific (breast, colon, prostate, brain, stomach, liver etc.) CSCs characterization is not discussed. Adding a table highlighting different tumor specific CSC biomarkers would be more informative.
  1. The authors should discuss the molecular/cellular signaling pathways regulating CSC stemness in detail. Adding a figure will greatly enhance the visibility of article. The authors should also debate the effect of tumor microenvironment on CSC stemness and treatment relapse. Clonal hematopoiesis of indeterminate potential predispose towards higher risk of developing hematological malignancies and disease relapse, and should be discussed.
  2. The flavonoids compound mediated regulatory pathways are not well described. The anti-CSCs effect of genistein, Broussoflavonol B, Icaritin and its analogue SNG1153, Morusin, Casticin should be discussed in detail. Adding the the anticancer mechanisms/mode of action of each compound in Table 1 will be more explanatory. Additionally, the authors should discuss the advantage and limitations associated with the clinical translation of these compounds.
  3. Role of dietary phytochemicals, curcumin, indole-3-carbinol, polyphenols, alkaloids etc in CSC management should also be briefly discussed. The authors should add a table describing natural compounds undergoing clinical trials that targets CSCs.

Author Response

Reviewer 3:

This review described the hallmark of cancer stem cells (CSC) and discussed the potential of natural products especially flavonoids, chloroquine and metformin in targeting multiple signaling pathways regulating cancer stem cell self-renewal, proliferation, survival, and their potential to augment the efficacy of standard cancer therapy. The review is timely, nicely written and very comprehensive.

We thank the reviewer for the positive assessment of our manuscript.

Major comments:

Though the authors discussed CD44 and CD133, ESA and ALDH, other CSC biomarkers including CD24, CD49, and CD90 etc. as well as biomarkers for tissue specific (breast, colon, prostate, brain, stomach, liver etc.) CSCs characterization is not discussed. Adding a table highlighting different tumor specific CSC biomarkers would be more informative.

This was done as per the reviewer’s request and now appears in Table 1 and in the text.

The authors should discuss the molecular/cellular signaling pathways regulating CSC stemness in detail. Adding a figure will greatly enhance the visibility of article.

A figure summarizing the relevant signaling pathways and the impact on NPs on these pathways was prepared as per the reviewer’s request and now appears as Figure 2.

Round 2

Reviewer 1 Report

All concerns addressed

Author Response

My reply is enclosed in the attached file.

Reviewer 2 Report

I have read with revised version of the article entitled “Flavonoids targeting cancer stem cells for augmenting cancer therapeutics”, and I thank the authors for improving the review.

  1. Figure 1 is a very nice Figure; however, it suggests that the effects of all flavonoids on CSCs are through the effect on their self-renewal. I do not think that is correct and I suggest that the authors should also include another Figure which would specifically classify and show the effects of discussed compounds on various hallmarks of CSC, e.g. ability to form spheres, self-renew undergo EMT or become drug resistant.
  2. The section of FDA approved drugs does not make sense in the new settings where the review is restricted to flavonoids only. They are not flavonoids, so why are they discussed in this review?
  3. The general part about the CSC biology seems to me a bit too simplistic and repetitive with other CSC-focused reviews. I would try to reformulate it a bit.

Author Response

(The authors gave the same response as above.)

Reviewer 3 Report

The authors have significantly modified the manuscript, which reads much better now.

In my opinion, the review would be more impactful if the authors also add a short note highlighting advantage/limitations of each flavonoid compound.

Author Response

(The authors gave the same response as above.)

Round 3

Reviewer 2 Report

I would like to thank the authors for their reply. As they mentioned I meant Figure2 but wrote Figure 1. Still, they did not modify it nor they introduced another Figure. I believe that Figure3 would be highly beneficial to include, depicting the influence of different compounds on different hallmarks of CSCs such as EMT capability/dissemination in the organism, drug resistance and tumorsphere formation or insensitivity to cell death including cell death elicited by the immune system.

In conclusion the authors mention twice that the drawback in flavonoids is the poor bioavailability, once is enough.

Author Response

As they mentioned I meant Figure2 but wrote Figure 1. Still, they did not modify it nor they introduced another Figure. I believe that Figure3 would be highly beneficial to include, depicting the influence of different compounds on different hallmarks of CSCs such as EMT capability/dissemination in the organism, drug resistance and tumorsphere formation or insensitivity to cell death including cell death elicited by the immune system.

Fig2 was modified and we added Fig3.

In conclusion the authors mention twice that the drawback in flavonoids is the poor bioavailability, once is enough.

Corrected